# Epileptic Seizure Detection Using Machine Learning: Taxonomy, Opportunities, and Challenges

**DOI:** 10.3390/diagnostics13061058

**Published:** 2023-03-10

**Authors:** Muhammad Shoaib Farooq, Aimen Zulfiqar, Shamyla Riaz

**Affiliations:** Department of Computer Science, University of Management and Technology, Lahore 54000, Pakistan

**Keywords:** epileptic seizures, epilepsy diagnosis, machine learning electroencephalogram (EEG), feature extraction, classification

## Abstract

Epilepsy is a life-threatening neurological brain disorder that gives rise to recurrent unprovoked seizures. It occurs due to abnormal chemical changes in our brains. For many years, studies have been conducted to support the automatic diagnosis of epileptic seizures for clinicians’ ease. For that, several studies entail machine learning methods for early predicting epileptic seizures. Mainly, feature extraction methods have been used to extract the right features from the EEG data generated by the EEG machine. Then various machine learning classifiers are used for the classification process. This study provides a systematic literature review of the feature selection process and classification performance. This review was limited to finding the most used feature extraction methods and the classifiers used for accurate classification of normal to epileptic seizures. The existing literature was examined from well-known repositories such as MDPI, IEEE Xplore, Wiley, Elsevier, ACM, Springer link, and others. Furthermore, a taxonomy was created that recapitulates the state-of-the-art used solutions for this problem. We also studied the nature of different benchmark and unbiased datasets and gave a rigorous analysis of the working of classifiers. Finally, we concluded the research by presenting the gaps, challenges, and opportunities that can further help researchers predict epileptic seizures.

## 1. Introduction

Epilepsy is a widespread neurological disorder that is common yet deadly if it goes untreated. This disorder affects people of all ages. A complex chemical change appears in the nerve cells of the brain for a seizure to occur. These chemical changes take place in the nerve cells that are made up of positive ions and negative ions that generate electrical signals [1]. These abrupt changes lead from mild jerks to severe, generalized, and prolonged convulsions. This neurological condition not only causes issues in movement but also disturbs the control of bowel or bladder function as well as affects consciousness and disturbs cognitive functions [2].

The populations that are affected by epilepsy or epileptic seizures are seventy percent adults and thirty percent children. The cause of epilepsy in adults and children in 70% of the cases is unknown. Clinical terminology states that if seizures are recurrent, they are termed epilepsy. It is stated that there are two classifications of seizures. One is partial, and the other is a generalized seizure [3]. In a partial seizure, also called a focal seizure, only a certain portion of the brain is damaged, whereas, in a generalized seizure, the whole brain is damaged. Mainly, epilepsy has four important stages, which include interictal, preictal, ictal, and postictal.

The most stereotyped way of detecting abnormal seizures is by manually using a device called an electroencephalogram (EEG). EEG is performed by placing an electrode on the scalp in a way that is either invasive or non-invasive [1]. It is acknowledged that epileptic seizure detection techniques differ under different conditions of the dataset across the electroencephalogram (EEG) [3]. Mainly, it satisfies the fact that there are different characteristics of EEG under different conditions. It is also very important to extract the right features for better prediction of an epileptic seizure. These techniques help with the accurate classification of data, hence leading to better accuracy of results.

There has been a great focus on EEG analysis for feature extraction, which has a very important role in the detection and prediction of various brain disorders [4]. Most of the research is carried out in differentiating normal seizures from epileptic seizures. Hypothetical testing for feature refinement of the feature selection method is used, and wavelet transforms for the improvement of classification [5]. The computational complexity of the classifier decreases with the right selection of features. Furthermore, studies are moving forward for the prediction of epileptic seizures, which is considered a more challenging problem [6,7].

Multiscale principle analysis and EEG decomposition methods are used along with feature extraction and selection methods which require statistical features, as well as empirical mode decomposition is used in decomposing the EEG signals [8]. Hence, it is highly challenging to detect seizure disease as well as discover whether the brain-related disease is effective. Machine learning (ML) classifiers are ideal for classifying the data of EEG and detecting seizures by revealing the associated patterns with high performance. A wide number of seizure detection approaches have been developed by using multiple machine learning classifiers as well as features. The selection of an appropriate classifier and feature is the major challenge [9]. Therefore, this study provides an analysis of the feature-extracting methodologies used for the detection of epileptic seizures along with the ML techniques.

Furthermore, the prime focus of this study is to identify the gaps and challenges and pinpoint opportunities and further advancements that lead the researchers to better opportunities in this area. This review contributes in four ways; firstly, we have identified the most used feature extraction methods, classifiers, and datasets for precise classification of epileptic seizures. Secondly, it presents a hierarchal taxonomy that recapitulates the state-of-the-art used solutions for seizure detection. Thirdly, the research results are concluded by presenting current challenges and gaps that researchers and people are facing in accurate seizure detection.

This paper is organized into five sections. Section 2 presents the background of epileptic seizure detection with ML approaches. Section 3 provides us with a research methodology where the research objectives, research scheme, research questions, screening and selection methods of the papers, and research string is generated and described in detail for researchers to get the gist of the whole paper. Section 4 gives the data analysis, which includes the search results in phases as well as a discussion and assessment of the research questions in detail. Section 5 presents a hierarchal taxonomy and discusses the issues and challenges that were faced in ML-based seizure detection. Section 6 concludes this whole research.

## 2. Background

Epilepsy is a continuing unprovoked seizure that changes the normal activities of the brain, which appears as an unpredicted electrical disturbance of the human brain. This brain disturbance is recorded by using EEG for disease analysis that indicates the seizures. Therefore, multiple ML methods have been developed for the classification of EEG signals. The most commonly used ML classifiers are neural networks, decision trees, K-nearest neighbors, and support vector machines [10,11,12,13,14].

Potapov [15] has investigated that while extracting useful features and suppressing noise, the preprocessing of signals can lose the important information that is vital for classification. Since the classification accuracy affects initial classification data, therefore initial data is necessary to be taken into consideration for the signal classification method.

Harikumar et al. [16] implemented fuzzy logic by using a genetic algorithm for the classification of non-epileptic and epileptic signals. They measured the risk factor by designing multiple methods and binary genetic algorithms to measure the accurate risk for seizure detection.

Sharmila et al. [2] designed a framework to detect epileptic seizures from epileptic patients and non-epileptic subjects through EEG. The designed framework depends on the discrete wavelet transform to analyze EEG signals by using linear as well as non-linear classifiers. Furthermore, an epilepsy diagnostic scheme has been proposed based on bootstrap aggregating and tunable Q factor [17]. Moreover, Mursalin et al. [18] proposed a novel method for epilepsy seizure detection by implementing a feature selection method with a random forest.

There are multiple approaches for epilepsy seizure detection, including feature selection, feature extraction, and dimensionally reduction methods that have been developed based on fuzzy logic [19,20]. Hence, several studies deal with the EEG signals by performing different analyses for epileptic seizure detection.

Hence, several studies deal with the EEG signals by performing different analyses for epileptic seizure detection. However, finding an authentic technique to resolve all those issues is worthwhile as well as important. This review research identifies the number of studies presented in this field for epileptic seizure detection using ML techniques.

## 3. Research Methodology

This systematic literature review (SLR) provides the full procedure to collect and investigate the relevant articles from the selected studies. Figure 1 represents the SLR process that includes (1) research methodology: (i) research objectives, (ii) research strategy, (iii) research questions and motivation; (2) selection and screening of papers: (i) inclusion/exclusion criteria, (ii) data extraction analysis; (3) assessment of research questions: (i) justification of research questions.

### 3.1. Research Objectives (ROs)

The following are the primary objectives of this research:

RO1: Understanding brain signals during an epileptic seizure and differentiating normal seizure from an epileptic seizure.

RO2: An understanding of the performance of machine learning classifiers used for classification.

RO3: Features extraction techniques used for the correct epileptic seizure prediction.

RO4: Identifying gaps and challenges in the already published research as well as suggested future research directions.

### 3.2. Research Questions (RQs)

In order to accomplish a good SLR, research questions (RQs) were formed as the first step of this research. Furthermore, extensive research planning was carried out to accomplish this review research.

A selection and screening process was done in which the inclusion and exclusion criteria were described to further narrow down the research. In addition, the defined RQs are assessed and discussed thoroughly. Table 1 shows the RQs with their corresponding motivation.

### 3.3. Search Scheme

The most significant part of an SLR is to create a search strategy and execute that strategy in a systematic manner. Firstly, the goal is to collect the most relevant articles based on the chosen domain. The procedure further requires an illustration of the search string, literature resources that are utilized for search application, and the inclusion/exclusion criteria strategy to obtain the most significant and relevant articles from the pool of articles.

#### 3.3.1. Search String

A keyword-based search string was formulated in order to conduct an effective search to gather relevant studies by using five well-known online repositories. Keywords used for finding the relevant articles, as well as their alternate words, are described in Table 2. The “+” sign indicates the inclusion criteria for studies that have said terms.

To form a search string the logical operators “AND” and “OR” were used as a combination of a finalized alternate keyword to form a search string. The operator “OR” is the indication of additional options for the research, and the “AND” operator is used for joining the terms in order to form the relevant search terms for relevant results. The search string that is finalized contains four fragments. The first fragment is utilized to obtain the results that are related to machine learning, and the second fragment looks for results that include epileptic seizure. However, the third fragment includes results related to feature selection methods, and the last fragment shows detection.
(1)$$R=\forall [\left(ML\vee SL\vee DL\right)\wedge \left(ES\vee C\vee E\right)\wedge \left(FE\vee FS\right)\wedge \left(D\vee D\vee C\right)]$$

In the above Equation (1), R represents the search results while ‘∀’ represents ‘for all’, ‘∨’ depicts ‘OR’ operator and ‘∧’ sign used to indicate ‘AND’ operator and combining these search terms formulate the search string that is expressed in Table 2. Generically, the search term in equation (1) can be expressed as:

((machine learning OR supervised learning OR deep learning) AND (“epileptic seizure” OR “convulsions” OR “epilepsy”) AND (“feature extraction” OR “feature selection methods”) AND (“diagnosis” OR “detection” OR “classification”))

#### 3.3.2. Literature Resources

The journals that were selected for executing this research were well-known journals and were selected from their online repositories. Their names and details are mentioned in Table 3.

#### 3.3.3. Inclusion and Exclusion Criteria

Parameters defined for inclusion criteria (IC) are:
IC (1)Include studies that were primarily conducted for epileptic seizure prediction using machine learning techniques.IC (2)Feature extraction methods targeting wavelet transform methods for the decomposition of EGG signals.IC (3)Studies that encompass machine learning classifiers that included RF, SVM, ANN, and KNN.

The exclusion criteria applied to all the articles to exclude out-of-scope articles such as:
EC (1)If the study did not involve any feature extraction techniques that involve wavelet transform techniques.EC (2)The studies did not involve the classifiers such as RF, SVM, ANN, and KNN.

### 3.4. Selection of Relevant Papers

In order to stay relevant to the topic, research from 2017 to 2022 was selected for this paper. Various articles were selected based on the title and relevance to the said topic. As the process proceeded, the duplicate papers and some papers that did not meet the criteria of research were eliminated from the list of papers. Firstly, the studies were filtered based on the titles, and duplicate papers were removed. There was a bundle of irrelevant papers that were not related to the domain. In the selection process, a careful review of the abstract was given, and articles were included that described the prediction methods of epileptic seizure along with the feature extraction and classification techniques. Furthermore, the inclusion and exclusion criteria were implemented to refine the articles.

### 3.5. Abstract-Based Keywording

In the abstract-based keywording method, the abstract foregoes a thorough analysis in order to perceive the main idea of the article as well as to discover the most relevant keywords. Furthermore, the identified keywords were combined due to the understanding of the research contribution in the domain. The main keywords that were chosen because of their direct importance with epileptic seizures were, (ML), (EEG), “Feature extraction” (FE), and “Automated Epilepsy Seizure Detection” (AESD).

## 4. Data Analysis

In this section, a clear step-by-step study selection process of the selected articles is described in phases. It describes how the chosen articles were analyzed thoroughly in order to answer the research questions.

### 4.1. Search Results

In the study of epileptic seizure detection, there are a few steps that are shown in Figure 2 in order to carry out this review research. Different well-known data sources and datasets were collected and studied, along with feature extraction and classification methods. Initially, a total of 48 research papers were collected online from different data sources. The search was conducted in four different phases. In phase 1, a title-based selection was made where the titles of the collected research papers were assessed from the 48 articles. In phase II, duplicate, irrelevant articles were removed based on the inclusion and exclusion criteria.

In phase III, abstract-based keywording was applied, where the abstracts of the remaining articles were read and assessed thoroughly, which left us with 36 papers. Furthermore, in phase IV, a full-text base analysis was done, and the systematic literature review was conducted and proceeded on a total of 25 papers. Figure 3 shows the digital library-wise article selection process, which shows that articles were selected from well-known repositories online. In the end, the results of SLR questions are presented in classification Table 4.

### 4.2. Assessment of Research Questions

In this section, the selected 25 articles were analyzed and assessed to explain each research question comprehensively. The facts obtained regarding defined research questions are discussed and assessed in this section.

#### 4.2.1. Assessment of Question 1: What Machine Learning Classifiers Are Used in the Majority of the Research for the Diagnosis of Epileptic Seizure?

In this research question, machine learning classifiers are discussed that are used in the detection of epileptic seizures. The most commonly used classifiers in the literature are identified and discussed in this research question.

##### Random Forest (RF)

This approach focused on the performance of the selected classifiers in the detection process of epileptic seizures. Wavelet packet features technique is used and adopted random forest classifier as the epilepsy state classifier [34]. They have adopted a feature-based splitting method for the tree nodes as well as generated a decision tree for each dataset. Splitting features were selected based on the criteria of gain. Classification accuracy obtained by random forest was 85% based on the number of decision trees provided, which ranged from 50 to 1200. However, this classification was acquired in the pre-ictal stage while ictal and inter-ictal stages achieved 97% and 98%, respectively, and evidently performed better than other classifiers. Moreover, it used an MSC approach which is called a multistage state classifier based on a random forest algorithm [37]. The structure of MSC includes three random forest classifiers that contain the basic logical decision thresholds which control the internal state transitions.

The study followed cross-frequency coupling and continuous wavelet transform for feature extraction and later used the MSC model for classification. The MSC model was trained, and optimization of two parameters was done through multi-iteration and 5-fold ROC cross-validation over the training set. First, in each ensemble random forest, the optimal number of estimators was found, and later the optimal value for the threshold was set. The testing of the model was done with an interictal time of 10 min and an ictal period of 66 s. The performance of the model was assessed based on two groups, one which gained training and one with no training. It was stated that their model performed well with the first group with an overall accuracy of 95%, group 2 achieved 79% accuracy, and all patient sets achieved an overall 82% accuracy by using a random forest algorithm-based model for epileptic seizure detection.

Since random forest is considered a robust technique in selecting large features, therefore, it is widely used not only for classification but also for feature-extracting methods for the better analysis of EEG signals in order to detect epileptic seizures [18]. Random forest was used in this article for the selection of the top features in which out of 178 features, 20 variables were chosen, and they were forwarded to ANN for further classification. Table 5 presents the RF technique applied in the literature for epilepsy seizure detection.

##### Support Vector Machine (SVM)

Support vector machine (SVM) is a technique in machine learning which is used widely and performs excellently in the areas where classification, prediction, estimation, regression, and forecasting are involved [10]. A study proposed the SDI method and used SVM for classification. They used a radial basis kernel function (RBF) and determined hyperparameters by using the optimization technique. The class labels 0 and 1 were used to train the classifier for normal and epileptic seizure in EEG signals, respectively. They provided training for each database separately by utilizing the method of cross-validation and the leave-one-subject-out method. A comparison was made with other linear approaches, such as linear SVM, linear regression, and linear discrimination. It was noted that SVM with (RBF) performed more excellently than the compared classifiers giving a sensitivity of 97.3%, a false detection rate of 0.4/h, an F score of 97.22%, and a median detection delay of 1.5 s. Bhattacharya et al. [13] proposed a technique where they utilized the empirical wavelet transform (EWT) method and decomposed the signal into rhythms, and further used the fast Fourier transform (FFT) in order to determine the signal’s frequency components. Wavelet functions are scaled at each segment, and then the sub-band signals are reconstructed by using the EEG rhythms.

The least-square support vector machine is used in this study, where the signal of EEG is classified into focal and non-focal signals. A total of 50 pairs of focal and non-focal EEG signals were used here. They also used the same method on 750 signal pairs which led to a classification accuracy of 82.53%, sensitivity of 81.60%, and specificity of 83.46%.

Subasi et al. [23] used a hybrid SVM model in which the SVM has kernel-type parameters and regularization constant C, which leaves an impact on the performance. Parameter values are either default values, or they are values that are selected manually through trial and error. GA-SVM and PSO-SVM algorithms are used for the selection of these methods in this study for better classification techniques in epileptic seizure detection problems. However, the hybrid SVM classifier, due to the longer time for parameter selection, could not outperform GA-SVM and PSO-SVM, where PSO-SVM performed slightly better than the rest achieving an accuracy of 99.38% and GA-SVM achieved 98.75% while SVM achieved 97.87% accuracy. Table 6 presents the SVM approach used in the literature for seizure detection.

##### K-Nearest Neighbor (KNN)

KNN classifier is based on learning by analogy. It searches for the pattern space neighbors that are closest to a given unknown sample. Closeness is defined in terms of distance. The unknown sample is assigned the most common class among its neighbors [11]. The KNN classifier is used for the detection of any abnormality from the EEG signal during the epileptic seizure after the features are extracted from different levels of decomposition [28].

The average accuracy obtained by the KNN classifier that is extracted from the wavelet decomposition with sy4 wavelet at level 5 was 97.50% for the detection of an epileptic seizure. A new technique for the detection of epileptic seizures has also been developed in which they used statistical features which were obtained by the discrete wavelet transform and feature reduction techniques. The techniques include PCA and LDA for the classification of normal EEG signals with epileptic seizure EEG signals by using k-NN and naive Bayes classifiers [31]. The proposed method has shown high accuracy of classification by using the LDA method for feature reduction and the KNN method for the Bonn University database. Table 7 presents KNN performance metrics applied in the literature for epilepsy seizure detection.

##### Artificial Neural Network (ANN)

The detection of an epileptic seizure by the use of EEG signals is discussed in order to achieve the right approach to obtain classification accuracy of normal and epileptic seizures [25]. For that, this study has used the discrete wavelet transform (DWT) feature extraction as well as GA-ANN for the process of selecting features that are more effective along with the intended results. It has an enhanced accuracy measure which can be achieved in two classes, including epileptic seizure and normal, and three class classifications, including epileptic seizure, normal, and seizure-free. The 5-level decomposition method with db4 DWT was used. Also, ANN and SVM contributed to providing an accurate classification of 100% and 98.7%, respectively, with this method. Table 8 presents ANN applied in the literature for epilepsy seizure detection.

A novel approach is used for the EEG signal diagnosis for epileptic seizure by using the multi-DWT as well as a genetic algorithm that is used with the four classification methods, such as SVM, ANN, NB, and KNN [28]. The results of this study showed that ANN outperformed other classifiers with this technique. The EEG signals are firstly preprocessed, which is the primary step of the method that helps in increasing the performance of the system and in removing noises. The proposed system foregoes various stages for detecting epileptic seizures. Feature extraction is the second step that ends up generating a features matrix that is later used in the classification process of EEG. The verification of success is done by the implementation of datasets in 14 combinations. The results were measured in terms of accuracy, specificity, and sensitivity, and it was noted that with this technique, ANN performed relatively better than the other classifiers. The accuracy achieved by ANN was 97.82%, while SVM, NB, and KNN achieved 97.15%, 97.32%, and 97.58%, respectively.

Saric et al. [29] trained the neural network algorithm and then tested it on 822 signals from the database. Five crucial features were used as input extracted from the EEG signals in time-frequency analysis and continuous wavelet transform, as well as subsequent statistical analysis. A total of 583 which is 70% of the samples out of the total samples, were used for the system development, and 239, which is 30%, were used for testing the performance of the proposed model. The right parameters for ANN were selected by k-fold cross-validation. Finally, the ANN classifier was implemented on the proposed model, and according to the study, the results showed that high accuracy of 95.14% was achieved with ANN implemented on this model.

Domain importance was defined by seeing a gradual increase in interest in the said study field. Domain importance can be measured by gradually increased interest in the selected field of study. Table 9 shows the most commonly used and selected ML classifiers for this review. The published articles were selected in the years ranging from 2017 to 2022. Figure 4 presents a yearly distribution chart of selected articles to the best of our knowledge, showing a non-linear increase and decrease of domain per year. In the end, a total of 25 papers were selected; one paper was from 2017, four papers were from 2018 and 2019, as well as 11 papers in 2020. However, there is a gap seen in 2021 and 2022, where only four and one paper were identified, respectively. Overall, it is shown that there is a rise and fall in the study in increasing years.

#### 4.2.2. Assessment of Question 2: What Kind of Features Extracting Methods Are Being Used, and What Features Are Being Extracted from the Eeg Signal?

This research question describes the feature extraction techniques that are commonly used in the literature—Table 10 feature extraction methods used in the literature with different ML approaches for epileptic seizure detection. A multi-view deep feature extraction technique has been proposed for epileptic seizure detection in real-time [38]. A model is introduced by the authors in which the EEG signal is first segmented by a fixed sliding window across the whole signal and passed two parameters which are window length l and window step s. It is stated that by increasing the length of the signal, the accuracy of the recognition process increases, which causes delays in real-time applications. Therefore, they fixed the length of the sliding window l to 3 s and step s to 1 s. The study considered the time-frequency domain and used the short-time Fourier transform (STFT) to obtain spectrogram representation by converting the EEG signals through STFT.

For feature extraction, as soon as the spectrograms are generated, the proposed model can extract a set of deep features automatically by intra-correlation and inter-correlation of the EEG channels, which includes cross-channel features and intra-channel features. For instance, given a single-channel EEG spectrogram, we can regard it as a spectral image where various spatial features are extracted within each fragment. In this step, high-dimensional raw features are integrated into low-dimensional latent characteristics with meaningful interpretation. An SSDA-based channel selection method is also introduced in order to filter the irrelevant features and to select channels that can extract critical features such as intra-channel features so the classification process can be more efficient. In this study, the accuracy obtained was 98.97% considering the approach used.

Furthermore, empirical wavelet transform (EWT) is utilized, and adaptive wavelets are constructed to extract different modes of the EEG signal [59]. For instance, the Fourier transform is used to extract the frequency components. Further, proper segmentation of the Fourier segment is done in order to extract these modes. Moreover, the wavelet and scaling coefficients are collected through their correspondence to each segment, along with the reconstruction of sub-bands. The study primarily focused on focal and non-focal seizure by extracting the rhythm of the EEG signals using EWT. The features extracted in this study are non-linear. Thus, the area is generated using the central tendency measure (CTM) generated by the 2D reconstructed phase space plot (RPS), giving an accuracy of 90%. Moreover, this study utilized DWT with arithmetic coding for feature extraction [6]. They used db4 wavelet as an appropriate technique for non-stationary signals which is very effective in detecting sudden spikes that determine epileptic EEG signals.

Harpale et al. [4] used two types of statistical feature extraction methods, that is, time and frequency domain statistical feature extraction and pattern adaptive wavelet transform feature extraction method. The methods of time and frequency domain are the artifacts from the scalp EEG signal that are removed using the independent component analysis. The extracted features obtained were coefficient variance, mean & variance, root mean square, kurtosis, power, the sum of mean, power spectral density, and zero crossing rate. They considered 512 samples as the window size per rectangular window without the overlap with a total of 23 channels. Furthermore, the features that are extracted from these channels are used to form the final features by averaging them with the same size as the window. The final features are further grouped into signal frames that are normal and pre-ictal. It is about 30 s before the seizure starts, and seizure frames throughout the training phase. The final features of the signal calculated during the testing phase are done just like how they were calculated in the training phase. 

Moreover, in the pattern-adapted wavelet-based feature extraction method, a normal EEG signal-based wavelet transform is constructed, which has a length of 36 that is centered at 256 samples of the normal signal. Further, a pattern-adapted wavelet is applied to the signal to estimate the position and scale of the pattern so that absolute values of the wavelet coefficient can be formed. The duration of the seizure is calculated as seizure duration = scale * sampling period and centered time = index * sampling period. Therefore, a feature vector is constructed, and seizures are detected along with root mean square, power spectral density, and standard deviation are calculated as the extracted features.

Hussain et al. [26] proposed a feature-extracting strategy that is multimodal and used for the detection of epileptic seizures. We know how the complexity of EEG signals arises in terms of non-linear and non-stationary behavior. This study has extracted the features which are situated on the time domain, frequency domain, complexity-based measures, and wavelet entropy methods that are used in the classification process of healthy subjects and subjects that are foregoing an epileptic seizure heart rate oscillations. The study also extracted non-linear features by using sample entropy which is based on the KD tree algorithmic approach (fast sample entropy) as well as approximate entropy.

#### 4.2.3. Assessment of Question 3: What Are the Gaps and Challenges in the Detection of Epileptic Seizures?

In this research question, we have discussed the current challenges and gaps that users face during the detection of epileptic seizures. The major gap that has been witnessed is that the techniques used majorly for generalized epileptic seizure detection can be utilized to work solely on petit mal epilepsy. To the best of our knowledge, limited data was observed on the detection of petit mal epilepsy. Usually, absence seizures have a 3 Hz per second spike-wave discharge pattern and often go unnoticed in kids because they occur for a short duration, usually for 30 s [60]. However, they are very fast and may occur 10 to 30 times a day, and are caused only in children, mostly females. Surely, petit mal epilepsy begins at the time of childhood, but sometimes it disappears before puberty. However, it is important to study and analyze the behavior of children around the age of 4 to 14 because the absence of seizures can affect a child’s daily life, and it can become harder for them to cope with. The symptoms of this type of seizure include the child staring blankly and having no idea of their environment. Further challenges include that a large amount of dataset is required for validating the detection process of machine learning and deep learning techniques. To the best of our knowledge, most datasets do not acquire a larger sample of EEG signals and contain signals in chunks, and are deemed unsuitable.

#### 4.2.4. Assessment of Question 4: What Datasets Have Been Used in the Majority of the Research for Epileptic Seizure Detection?

Several studies considered the analysis of EEG signals on a publically available dataset from the University of Bonn [6,27,31,32,61,62]. The dataset consists of five subsets labeled A, B, C, D, and E. A total of 100 single-channel EEG segments are used in this dataset that contains a length of 23.6 s and 4097 total samples per channel. Set A and B were the labels of EEG recordings of five healthy subjects with open and closed eyes, respectively. Datasets C and D contained patients who have epilepsy but are not foregoing a seizure currently. The EEG recordings of C included the hippocampal formation from the hemisphere opposite to the epileptogenic zone, and dataset D included the records of the epileptogenic zone where the seizure usually arises. These recordings were collected at a time when the patients were not foregoing seizures. Set E contains the collection of patients that are going through the seizure activity recorded by the hippocampal focus. These datasets were recorded by using 128 channels. Channels that consisted of pathological activities were removed from the computation. The eye movement artifacts were also removed from the scalp, EEG sets A and B. The data was acquired by utilizing a 12-bit analog-to-digital converter with a 173.61 Hz sampling frequency. Further, they applied a bandpass filter to raw EEG data. EEG data consisted of five classes × 100 observations per class × 4097 (23.6 sec per observation).

However, Rabby et al. [35] utilized the same dataset but with spectral bandwidth that ranged from 0.5 Hz to 0.85 Hz of the EEG acquisition system, and the sampling of data points was done at 173.61 Hz and passed to a 40 Hz low pass filter. The dataset consisted of text files that were classified in binary as 0 or 1, meaning non-seizure and epileptic seizure, respectively. With this data, two matrices were generated containing the sampled signal data Dataset from Children’s Hospital called CHB-MIT EEG [4,9,30,36,38]. They recorded 22 patients at various times and made 654 files with uncontrollable seizures that cannot be controlled by medicines. The files consist of recordings that start from 1 h to 4 h with 16-bit resolution and a 256 sampling rate. This study included one female patient and five male patients aged from 1.5 to 22 years old. They used a standard international 10–20 system for the placement of sensors on the scalp and used 23 channels for the recording of the data. They carried out the experiment with a 23/256 duration of signal samples in each file. The first out of three datasets where the EEG recordings were collected were from the Institute of Neuroscience, India [24]. It included 19 channel scalp EEG, which included a duration of 58 h, taken from 115 patients, which included 67 males and 48 females whose ages ranged from 2.5 to 75 years old. Out of 115 subjects, there were 38 subjects suffering from epilepsy, and 77 were healthy. The international 10–20 system of electrode placement, along with the sampling rate of 128 Hz, was used to collect the EEG signals. The second database was from CHB-MIT from the Physionet repository, which is a publically available dataset. This contained 23 patients whose recording was made at a sampling rate of 256 Hz which included 844 h of data. The international 10–20 system bipolar montage was used on this dataset. The final database used in this study was obtained from TUH EEG, which consisted of 316 patients following the same international 10–20 electrode placement system with a 250 Hz sampling rate. From the EEG recordings, 222 out of 316 epileptic seizures were considered.

The epileptic EEG data was recorded of 16 subjects in Katip Celebi University, School of Management and neurology department by using surface electrodes [33]. A neuro fax device was used to record the EEG data from various channels with 100 Hz as the sampling frequency. Electrodes were placed with regard to the international 10–20 system. Maximum 2 epochs were used on each patient in this study, where each epoch contained 10 channels for 1 min with a total of 32 epochs. Furthermore, a long-term EEG recording of 275 patients was used from a public database called the European Union-funded database [3]. Moreover, the EPILEPSIAE database was used by the researchers, which is a widespread electroencephalography database of epilepsy patients.

## 5. Discussions

In this systematic literature review, we have investigated the implementation of multiple ML approaches for epileptic seizure detection. The investigated studies in this research indicate ML algorithms such as SVM, RF, ANN, and KNN are ideal for processing the dataset (BONN, CHB-MIT, Kaggle, and Fribourg) of the brain for epileptic seizure detection [25,26,27,28,29,30,31,32,33,34,35,36,37,38,39]. Nevertheless, each algorithm and approach has its own pros and cons. For example, SVM is an ideal approach for binary classification. Compared to KNN and ANN, the SVM approach has good detection accuracy. On the other hand, the performance evaluation of KNN is low, but KNN can handle massive dimensional data sets [57]. Furthermore, in conventional algorithms, ML approaches face difficulty in understanding the prediction outcome, and it is not possible for them to explain the patterns as well as hidden rules inside the model. Hence, they are effective and recommended for extracting useful information from chosen datasets. On the other side, advanced ML approaches such as LSTM, CNN, or RNN help in the extraction of different high-dimensional features, which is not possible with conventional techniques. Some state-of-the-art studies identified and analyzed that the time domain feature extraction approach is best with different statistical methods such as entropy, energy, skewness, mean, mode, etc. [63]. 

Moreover, kernel principal component analysis (KPCA) is also best for epileptic seizure disease classification on the basis of EEG signals [64]. Hence, KPCA is the best encoding approach with the nonlinear manifold method. This approach has been used widely in many datasets, such as facial pictures, health data, and sensor data.

In this section, we have presented the hierarchical representation of used ML techniques and data sets and discussed challenges and gaps in the respective domain.

### 5.1. Hierarchal Representation of Epileptic Seizure Detection Techniques

To summarize the discussion of the review, we have presented a hierarchal representation of used ML techniques, classifiers, and data sets in Figure 5. The problem of epileptic seizure detection was addressed as how it is disturbing the daily life of people [7]. Furthermore, it being a neurological disorder necessitates the early detection of the disorder in order to prevent great harm and sometimes death [8]. In this study, various feature extraction methods have been discussed, along with classifiers and datasets that are being utilized in the detection of an epileptic seizure. To the best of our knowledge, it has been shown that SVM, MLP, NB, DNN, ANN, and KNN are mainly used in various studies, along with feature extraction techniques for better detection of EEG signals on various datasets.

Furthermore, the datasets used in the majority are described along with their descriptions where most of the results vary with the variation of datasets picked for this problem [2,8,18]. For this review, many wavelet-based and similar techniques were targeted and used in the signal decomposition of the brain signal in order to understand various states of epileptic seizure and their prevention methods.

### 5.2. Challenges and Gaps

Researchers have the opportunity to predict epileptic seizure with feature extraction techniques by studying the non-linear features thoroughly and understanding their results on different classifiers. Furthermore, a variety of datasets is used, but due to different parameters for feature extraction methods, it becomes difficult to gain insight into a larger dataset with a combination of feature extraction techniques.Furthermore, this review research has been conducted to understand the basis of how epileptic seizure detection methods are being used in the domain of machine learning. Hence, for future work, researchers can work specifically on petit mal, known as absence seizure. Their detection is somewhat challenging because of its minimal duration and negligible visual symptoms. Whereas the occurrence of these seizures is more frequent and can psychologically affect a child’s life since it only occurs in kids from age 4 to 14.To the best of our knowledge, there has not been much literature regarding the ML techniques being applied solely for the detection of absence seizures. The major opportunity could be in the generation of a device that is user-friendly for a home environment and not as terrifying as an electroencephalogram for children.Normally, children with absence seizures show mild signs or symptoms of absence or abnormal behavior, such as staring blankly at the wall or not understanding a word someone says. A device should be made after testing the algorithms for the detection process with less chance of false positives so that the absence seizures can be detected and monitored at home by parents or guardians.

## 6. Conclusions

This study conducted a systematic literature review (SLR) on the detection of epileptic seizures, which provides an analysis of the papers selected for this research in the field of epileptic seizure detection methods. An analysis of ML classifiers was done along with the feature extraction methods being used in the study, and the data sources were thoroughly mentioned in the paper. Different datasets that are publicly available were seen and investigated, and most of the selected studies have used these datasets in their research. Feature extraction approaches were mainly focused on techniques that used wavelet transform, and signal decomposition was done for the prediction of an epileptic seizure. The classifiers studied were SVM, RF, KNN, and ANN, which showed good results while using these classifiers with the feature extraction methods. Furthermore, it is suggested to study the most relevant predictive models in the future to perform quality research along with a suggestion on the absence of epilepsy in children and generate a separate dataset for this type of epileptic seizure.

## Figures and Tables

**Figure 1 diagnostics-13-01058-f001:**
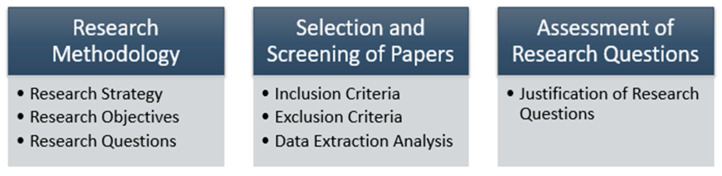
SLR Process Representation.

**Figure 2 diagnostics-13-01058-f002:**
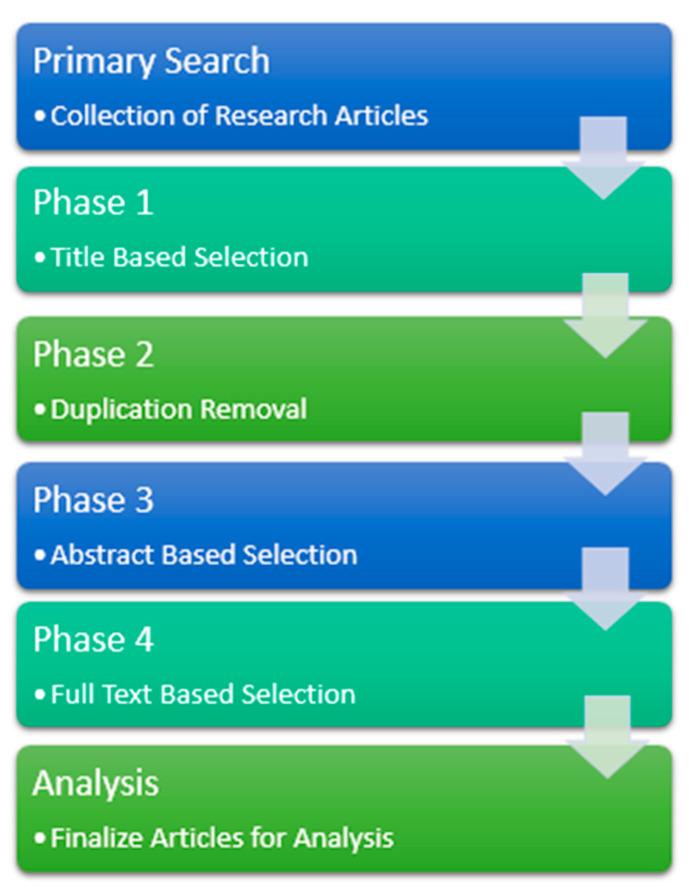
Selection Procedure.

**Figure 3 diagnostics-13-01058-f003:**
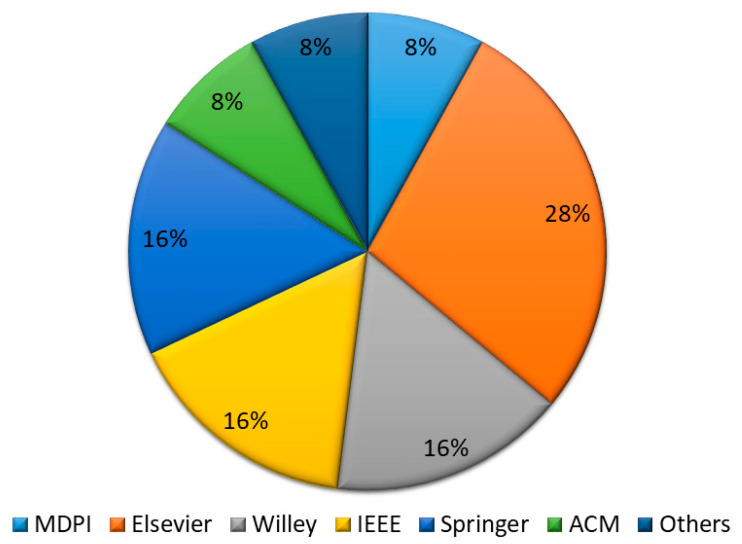
Selected studies repository ratio.

**Figure 4 diagnostics-13-01058-f004:**
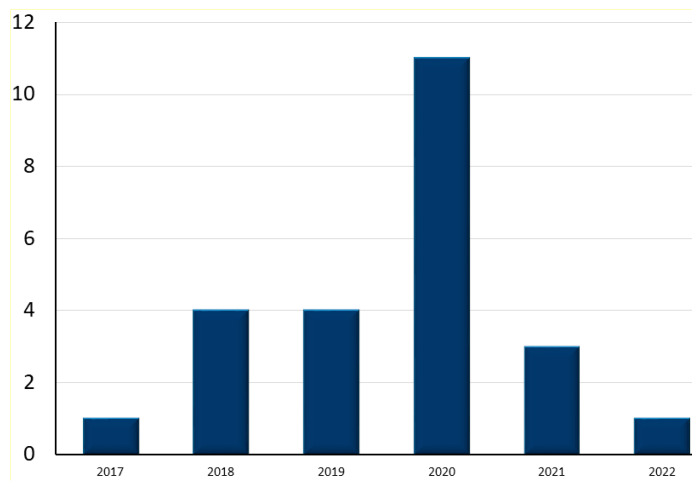
Publication frequency of studies based on selected years.

**Figure 5 diagnostics-13-01058-f005:**
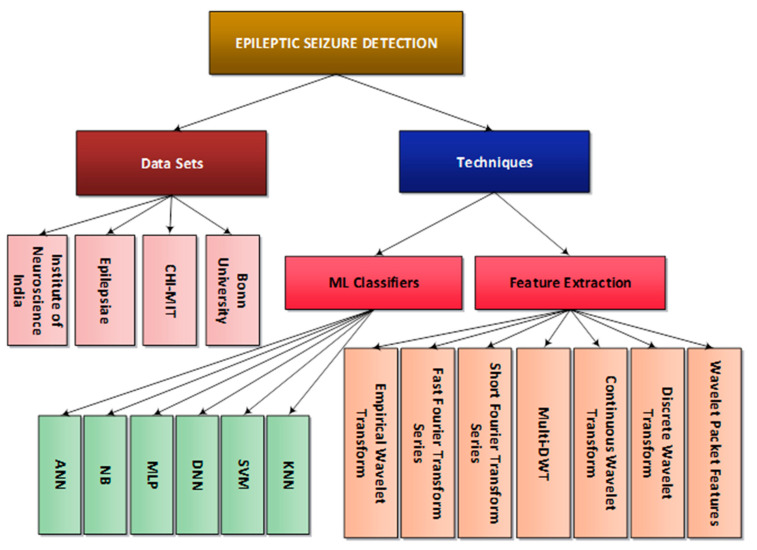
A hierarchal Taxonomy of Epileptic Seizure Detection Techniques.

**Table 1 diagnostics-13-01058-t001:** Research Questions and the motivation behind them.

	Research Questions	Motivation
RQ1	What machine learning classifiers are used in the majority of the research for the diagnostics of epileptic seizure?	For the understanding of the model requirements for the prediction of epileptic seizure
RQ2	What kind of feature-extracting methods are being used, and what kind of features are being extracted from the EEG signal?	To understand the performance of the classification process based on the features extracted
RQ3	What are the gaps and challenges in the detection of epileptic seizures?	This question aims to identify strengths and limitations in pattern recognition techniques and the performance of classifiers based on feature extraction on various datasets.
RQ4	What datasets are used for epileptic seizure detection?	To check the biases of the dataset

**Table 2 diagnostics-13-01058-t002:** Terms and keywords used in the search.

Terms (Keywords)	Synonyms/Alternate Keywords
**+Machine Learning**	Classification Techniques, ML classifiers, Classification
**+Epileptic Seizure**	Seizure detection, Epilepsy detection, Convulsions, Epileptic seizure detection
**+Feature Extraction**	-

**Table 3 diagnostics-13-01058-t003:** Publisher-wise search strings.

Data Repository	Relevant Search Strings
Science Direct	((Machine learning or supervised learning or deeplearning) and (“epileptic seizure” or “convulsions” or “epilepsy”) and (feature extraction or feature selection methods) and (diagnosis or detection or classification))
Springer link	((machine learning or supervised learning or deeplearning) and (“epileptic seizure” or “convulsions” or “epilepsy”) and (feature extraction or feature selection methods) and (diagnosis or detection or classification)[fields] [all fields])
Ieee Xplore	(((((((((((“all metadata”:”machine learning”) or “all metadata”:supervised learning) or “all metadata”:deep learning) and “all metadata”:epileptic seizure) or “all metadata”:convulsions) or “all metadata”:epilepsy) and “all metadata”:feature extraction) or “all metadata”:feature selection method) and “all metadata”:diagnosis) or “all metadata”:detection) or “all metadata”:classification)
MDPI	(“machine learning”[all fields] or “supervised learning” [all fields] or “deep learning”[all fields]) and (“epileptic seizures”[all fields] or “convulsions”[all fields] or “epilepsy”[all fields] or “feature extraction”[all fields] or “feature selection methods” [all fields]) and (“diagnosis”[all fields] OR “detection”[all fields] or “classification”[all fields])
WILEY	((machine learning or supervised learning or deep learning) and (“epileptic seizure” or “convulsions” or “epilepsy”) and (feature extraction or feature selection methods) and (diagnosis or detection or classification))
ACM Digital Library	((((“machine” or “supervised learning”) and learning) or deep learning or (“machine” and “learning”) and ((“epileptic seizure” or “convulsions”) and (“feature “ or “feature extraction”) and selection) or ((“feature extraction method”) and classification) or (“detection of” and (“classification” or “diagnosis”))))

**Table 4 diagnostics-13-01058-t004:** Classification Table.

Ref:	Year	Problem Tackled	Classifiers	Technique	Findings	Datasets
[1]	2020	Differentiating normal EEG signals with Epileptic seizure signals in ictal and inter-ictal stage	Linear and Non-linear ML techniques NB, KNN, MLP, SVM	CAD-based diagnoses using DWT, Wavelet Decomposition, Feature computation, and classification, Arithmetic Coding	Overall 100% accuracy	Bonn University
[4]	2021	Identifying pre-ictal and ictal state of EEG signals	Fuzzy classifier	Pattern adaptive wavelet transform	96% accuracy	CHB-MIT
[5]	2019	Misdiagnose in manual methods, so the procedure is automated	SVM KNN Deep Neural Networks	Feature Scaling Loss Function	SVM = 94% accuracy KNN = 74% accuracy	Bonn University
[9]	2022	Detecting seizure and non-seizure events	SVM RF	Tunable Q-wavelet transform	RF-sensitivity = 91.5% RF-accuracy = 93% SVM-sensitivity = 9 = 89.2% SVM-accuracy = 90.4%	CHB-MIT
[18]	2017	The automated framework created for the automated detection of epilepsy	ANN model RPROP+, RPROP-, SAG, SLR	Feature Selection Back Propagation	SLR = 99%, SAG = 97% in balanced classes and SLR = 87%, SAG = 89% in balanced classes	Preprocessed Epileptic seizure recognition on UCI repository. Balanced and Imbalanced classes.
[21]	2019	Seizure prediction and detection	Random Forest	DWT with 5-level decomposition	High classification sensitivity was achieved by this method, reaching 99.95% in comparison with other studies	Bonn University. Freiburg Hospital.
[22]	2018	Automated onset prediction	SVM MLP KNN RF	Multiscale principle analyzing for de-noising. EMD DWT Wavelet packet decomposition Inter vs. inter-ictal	With the Freiburg dataset and CHB-MIT dataset, high accuracy has been achieved, indicating both well-known datasets worked well with the used technique	Freiburg Hospital. CHB-MIT.
[23]	2019	A hybrid model for epileptic seizure prediction	SVM	PSO-based SVM GA-based SVM DWT with db4	SVM= 97.87% accuracy GA-SVM = 98.75% PSO-SVM = 99.38%	Publically available dataset
[24]	2020	Automated seizure detection process with a comparison of the proposed method with existing methods	SVM	Successive Decomposition Index SDI. Wavelet energy	The sensitivity achieved is 97.53% with F-measure with 97.22%	Ramaiah College Hospital. CHB-MIT. Temple Unit
[25]	2021	Detecting epileptic seizures and defining the right features	SVM ANN	DWT 5 level and statistical calculations. Statistical features were extracted	2-class: ANN = 100% accuracy, SVM = 100% accuracy 3-class: SVM = 98.7% accuracy ANN = 98.7% accuracy	Bonn University
[26]	2018	Monitoring of brain activity during an epileptic seizure and normal state	SVM KNN Decision trees	Time-frequency domain characteristics Non-linear wavelet-based entropy	SVM = 98% accuracy KNN = 94% accuracy	Bonn University
[27]	2020	Brain activity at different regions for timely and accurate detection of epileptic seizure	SVM KNN	Feature engineering (FT) Wavelet transform The sequential forward floating selection	SVM = 99%, 100%, and 100% in time, frequency, and time-frequency, respectively, KNN = 99.5%, 99%, 99.5%, respectively	Bonn University
[28]	2020	Classifying normal brain signals with an epileptic seizure while increasing accuracy and reducing the computational cost	ANN KNN NB SVM	54-DWT wavelets Derived features minimization by using Genetic Algorithm to select relevant features	ANN achieved higher accuracy, reaching 97.82% in comparison with the rest.	14 classification combinations using Bonn University dataset
[29]	2020	Effective real-time epilepsy diagnosis	Feedforward multi-layer neural network, MLP, ANN	Field programmable gate array solution (FPGA)	95% accuracy	TUH-EEG corpus database
[30]	2020	Predicting a seizure in a pre-ictal stage in terms of specificity and sensitivity	Deep learning techniques SVM CNN	FT EMD WT Feature extraction and handcrafted feature extraction methods.	Sensitivity = 92.7% Specificity = 90.8%	CHB-MIT
[31]	2018	Development of automated seizure detection	Random forest	Synthesizing generalized Stockwell transform (GST), singular value decomposition (SVD) based feature extraction. Changing n values for 4 cases to see if it affects the accuracies	Highest classification accuracies are 99.12%, 99.16%, 98.65%, 98.62% for four cases	Bonn University
[32]	2020	Checking performance in terms of accuracy to find relevant patterns related to different mental activities using feature extraction. Extracting features based on spectrogram.	K-means SVM Multilayer perceptron	STFT is used, and window parameters are set to obtain good results. K-means to extract features. Descriptors: Spectral peaks Frequency Time	The comparison was made with other works, and it was noted that the SVM kernel gave a better performance.	Bonn University
[33]	2020	Combining four different approaches to decompose non-linear and non-stationary signals into a finite number of oscillations (IMFs)	SVM KNN NB Logistic Regression	EMD and its DWT derivatives and use them to generate EEG into oscillations called IMF	EEMD provided better accuracies than EMD analysis. EEMD provided a robust feature extraction and results	Kahib-Celebi School of Medicine
[34]	2019	Pre-ictal stage prediction multi-class classification	Random forest	Wavelet packet features Wavelet packet decomposition	84% accuracy	CHB-MIT
[35]	2021	Detection of epileptic seizure	ANN, SVM, NN, CNN	Wavelet transform Singular value decomposition entropy Petrosian fractal dimension Higuchi fractal dimension	ANN outperformed other classifiers	Bonn University
[36]	2020	Reducing seizure frequency or prevention of epileptic seizure by early prediction	Random forest Decision trees	DWT Coefficient of variance with all sub-bands	99.81% accuracy	CHB-MIT
[37]	2018	Pre-clinical seizure state	Random forest	Cross-frequency coupling Multistate classifier	Sensitivity = 87.9% Specificity = 82.4% Area under ROC = 93.4%	Toronto Western Hospital Epilepsy Monitoring Unit
[38]	2020	Computationally efficient automated seizure detection	SVM	Successive decomposition Index	SDI = higher detection rate of epileptic seizure in terms of sensitivity, db1 = 97.53%, db2 = 97.28%, db3 = 95.80%, false-detection rate, db1 = 0.4/h, db2 = 0.57/h, db3 = 0.49/h median detection delay, db1 = 1.5 s, db2 = 1.7 s, db3 = 1.5 s f-measure db1 = 97.22%, db2 = 96.29%, db3 = 94.70%	CHB-MIT Ramaiah Medical College Temple University Hospital
[39]	2021	Presented an approach for the classification of EEG signals based on fuzzy classifier	FDT, Fuzzy random forest (FRF)	fuzzy classifier	99.3%	Bonn University

**Table 5 diagnostics-13-01058-t005:** A review on RF approach for seizure detection.

Ref:	Approaches	Feature Selection Methods	Datasets	Performance Metrics	Limitations	Accuracy
[40]	Random forest classifiers	L1-penalized robust regression	BONN, CHB-MIT	Class Acc	-	100
[22]	ANN, random forest, SVM, KNN	Power, mean, kurtosis, absolute mean std dev, skewness	CHB-MIT	Sen, spec, Acc	Time complexity	100
[41]	Forest CERN	9-statistical features	BONN, CHB-MIT	Class Acc	-	100
[42]	Random forest	IMF	Kaggle	Sen, spec, Acc	Sen, spec not mentioned	98.4
[43]	Random forest, SVM	Frequency, 10-time	UCI	ROC-AUC	-	98
[44]	Random forest, boosting, decision forest	Nine statistical features	Bern Barcelona	Pre, Rec, Fmeasure	High time complexity	96.67
[45]	Random forest	Time, frequency	EPILEPSY	Sensitivity	Spec not mentioned	93.8

**Table 6 diagnostics-13-01058-t006:** A review on SVM approach for seizure detection.

Ref:	Approaches	Feature Selection Methods	Datasets	Performance Metrics	Limitations	Accuracy
[46]	LS-SVM	DWT, FFT	Class Acc	BONN	High time	100
[47]	SVM	Energy	BONN, Barcelona	Class Acc	-	99.5
[48]	SVM	DWT	BONN	Class Acc	-	99.38
[49]	SVM	Time-frequency	CHB-MIT	Sen, spec	High time complexity	99.32
[50]	SVM	Time-frequency	CHB- MIT	Sensitivity (sen)	-	96
[51]	SVM	DWT	CHB-MIT	Avg	-	94.8
[14]	SVM	Permutation entropy	CHB-MIT	Pre, Rec, Fmeasure	Low prec and accuracy	93.55
[52]	SVM	DWT	BONN	Confusion Matrix	Low sen, pres	86.83

**Table 7 diagnostics-13-01058-t007:** A review on KNN approach for seizure detection.

Ref:	Approaches	Feature Selection Methods	Datasets	Performance Metrics	Limitations	Accuracy
[53]	KNN and GHE	-	BONN	Class Acc	-	100
[54]	Naive Bayes, KNN	Energy	EPILEPSY	Class Acc	-	98.75
[55]	KNN	Time-frequency	BernBarcelona	Sen, pre, NPR, ROC	NFR not mentioned	97.6
[56]	KNN	15-features	BONN	Acc, sen, spec	-	98
[57]	KNN	Genetic programming	BONN	Class Acc	Low accuracy	93.50
[58]	QDA, DT, KNN	Time-frequency	BONN	Sen, spec	Low sen, pres	85

**Table 8 diagnostics-13-01058-t008:** A review on ANN approach for seizure detection.

Ref:	Approaches	Feature Selection Methods	Datasets	Performance Metrics	Limitations	Accuracy
[50]	ANN	Time-frequency features	BONN	Pre, Rec, Fmeasure	-	100
[14]	ANN, random forest, SVM, KNN	Power, mean, kurtosis, absolute mean std dev, skewness	CHB-MIT	Sen, spec, Acc	Time complexity	100
[39]	QDA, DT, KNN	Time-frequency	BONN	Sen, spec	Low sen, pres	85
[12]	ANN	Line length feature	CHB-MIT	CHB-MIT	Low accuracy	52

**Table 9 diagnostics-13-01058-t009:** ML classifiers used in the selected articles.

Sr. No.	Classifiers	Used in
1	SVM	[1,10,13,14,15,16,33,34,41,42,46,47,49]
2	RF	[8,10,16,34,45,48,50,57]
3	ANN	[9,33,42,43,46,49]
4	KNN	[1,10,12,32,34,35,41,42,46,47]

**Table 10 diagnostics-13-01058-t010:** A review of recent research that applied **the** ML approach for seizure detection.

Ref:	ML Approaches	Feature Extraction Methods
[22,40,41,42,43,44,45]	Random Forest (RF)	L1-penalized robust regression, Power, mean, kurtosis, absolute mean std dev, skewness, 9-statistical features, DWT, entropy, IMF, Frequency, 10-time, Std, dev, energy, energy, STFT, mean, Nine statistical features, Time, frequency
[14,47,48,49,50,51,52]	Support Vector Machine (SVM)	DWT, FFT, Energy, Time-frequency, Permutation entropy
[53,58]	K- Nearest Neighbour (KNN)	Energy, 15-features, Time-frequency, Genetic programming, Time-frequency
[22]	Artificial Neural Network (ANN)	Time-frequency features, Power, mean, kurtosis, absolute mean std dev, skewness, Time-frequency, Line length feature

## Data Availability

This is a review research therefore data is available publically.

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
