# Peer review of "Epileptic Seizure Detection Using Machine Learning: Taxonomy, Opportunities, and Challenges"

_diagnostics, 2023, doi:10.3390/diagnostics13061058_

Round 1

Reviewer 1 Report

Citations must be numbered in the order of appearance in the text.

Chapter 2 (Research Methodology) has too many pages compared to its importance in writing a Review. In writing a Review, the main focus must be on the presentation of the field, aspects and technical instruments. That's why I think that chapter 3 is more important and that it should be developed.

There are other types of features on the basis of which EEG analysis can be done for Epileptic Seizure Detection. For example, measures of non-linearity, entropy, etc. Perhaps these should also be presented. Or if it is not possible, then the title of the work, the abstract and the introduction should emphasize the expectation that only works in which feature extraction was based on Wavelet and FFT are analyzed.

Author Response

Reviewer#1, Concern # 1: - Citations must be numbered in the order of appearance in the text.

Author response:  Thank you so much for your valuable suggestion. We have updated the reference sequence in the whole manuscript and presented it in the same order as it appeared in the text.

Author action: We have updated the references order in whole manuscripts including introduction section 1, Classification table 4, and section 3 (assessment of research questions) considered

Reviewer#1, Concern # 2: - Chapter 2 (Research Methodology) has too many pages compared to its importance in writing a Review. In writing a Review, the main focus must be on the presentation of the field, aspects and technical instruments. That's why I think that chapter 3 is more important and that it should be developed.

Author response:  Thank you for highlighting this point and giving your valuable suggestions. We have updated and developed section 4 (that was section 3) comprehensively. More studies have been identified based on ML approaches, feature extraction methods, datasets, performance metrics, limitations, and accuracy and included in section 4 in tabular form. Besides, classification table 4 has also been updated by adding more studies as well as by ordering the references.

Author action: We have updated the classification table 4 by adding more articles in section 4. Furthermore, in section 4 research question 1 has been updated by adding tables 5, 6, 7, and 8 related to different ML approaches including random forest, support vector machine, artificial neural networks, and K-nearest neighbor. Moreover, in research question 2 added table 10 by identifying feature extraction methods in different ML techniques.

Reviewer#1, Concern # 3: - There are other types of features on the basis of which EEG analysis can be done for Epileptic Seizure Detection. For example, measures of non-linearity, entropy, etc. Perhaps these should also be presented. Or if it is not possible, then the title of the work, the abstract and the introduction should emphasize the expectation that only works in which feature extraction was based on Wavelet and FFT are analyzed.

Author response:  Thank you so much for giving this valuable suggestion. We have added 27 new articles based on identifying their feature extraction methods. Table 10 presents different studies with their feature extraction methods entropy, measures of nonlinearity, and more.

Author action: We have added table 10 to the research question with multiple feature extraction methods. Apart from this, Tables 5, 6, 7, and 8 also present feature extraction methods individually.

Reviewer 2 Report

Comments and Suggestion for Authors:

1. In the introduction, the background of epilepsy and its detection methods are analyzed and introduced in detail, but there is no summary analysis of representative articles in this field, and it does not explain the current situation of related issues or the focus of controversy. It is hoped that the author can further summarize and generalize the previous research results, and explain the current situation of related issues or the focus of controversy.

 2. The author spends a lot of space introducing research methods and paper search methods in Research Methodology and Data Analysis. This part of the content is too redundant. It should be more streamlined and focus on the main research content of the article.

 3. In section 3.2.1-section3.2.4 When the author introduces various machine learning methods, feature extraction techniques and data sets, he simply lists and piles up previous methods and experimental conclusions, without comparing and analyzing different methods. I hope the author can summarize the characteristics of different research methods and analyze the differences between them.

 4. The content of the Discussion is too simple. The author just makes a simple classification of various research methods and technologies and does not summarize the various machine learning methods, feature extraction technologies and data sets proposed in this paper. It is hoped that the author can discuss the research methods presented in this article in more detail, and summarize the characteristics and advantages of different research methods and data sets.

 5. As a review article, the number of references in the literature feels too small.

Author Response

Reviewer#2, Concern # 1: - In the introduction, the background of epilepsy and its detection methods are analyzed and introduced in detail, but there is no summary analysis of representative articles in this field, and it does not explain the current situation of related issues or the focus of controversy. It is hoped that the author can further summarize and generalize the previous research results, and explain the current situation of related issues or the focus of controversy.

Author response:  Thank you for highlighting this point. We have added a new section in the manuscript namely the background section where we discussed the previous research results with their proposed solutions and limitations.

Author action: In section 2 we have discussed the background of epileptic seizure detection with proposed solutions and limitations.

Reviewer#2, Concern # 2: - The author spends a lot of space introducing research methods and paper search methods in Research Methodology and Data Analysis. This part of the content is too redundant. It should be more streamlined and focus on the main research content of the article.

Author response:  Thank you for this valuable suggestion. We have updated and developed section 4 (that was section 3) comprehensively. More studies have been identified based on ML approaches, feature extraction methods, datasets, performance metrics, limitations, and accuracy and included in section 4 in tabular form. Besides, classification table 4 has also been updated by adding more studies as well as by ordering the references.

Author action:  We have updated the classification table 4 by adding more articles in section 4. Furthermore, in section 4 research question 1 has been updated by adding tables 5, 6, 7, and 8 related to different ML approaches including random forest, support vector machine, artificial neural networks, and K-nearest neighbor. Moreover, in research question 2 added table 10 by identifying feature extraction methods in different ML techniques.

Reviewer#2, Concern # 3: - In section 3.2.1-section3.2.4 When the author introduces various machine learning methods, feature extraction techniques and data sets, he simply lists and piles up previous methods and experimental conclusions, without comparing and analyzing different methods. I hope the author can summarize the characteristics of different research methods and analyze the differences between them.

Author response:  Suggestion incorporated. We have discussed 4 ML techniques including random forest, support vector machine, artificial neural networks, and K-nearest neighbor in question 1 with their experimental results and accuracy level. However, more properties regarding experimental results have been discussed in Tables 5, 6, 7, and 8.

Author action:  We have updated question 1 by adding tables 5, 6, 7, and 8 in order to show the differences between the used approaches and proposed solutions.

Reviewer#2, Concern # 4: - The content of the Discussion is too simple. The author just makes a simple classification of various research methods and technologies and does not summarize the various machine learning methods, feature extraction technologies and data sets proposed in this paper. It is hoped that the author can discuss the research methods presented in this article in more detail, and summarize the characteristics and advantages of different research methods and data sets.

Author response:  suggestion incorporated. We have updated the discussion section by summarizing ML approaches and data sets and investigated which methods and data sets are ideal for epileptic seizure detection.

Author action:  Discussion section 5 has been updated by investigating and incorporating which approaches and data sets are ideal for epileptic seizure detection.

Reviewer#2, Concern # 5: - As a review article, the number of references in the literature feels too small.

Author response:  Thank you for highlighting this point. We have revised the manuscript and added 27 more references in the manuscript.

Author action:  2 references i.e. [68] and [69] have been added to classification table 4 however, 25 more references are added in tables 5, 6, 7, and 8.

Reviewer 3 Report

This work is very interesting. This review may be useful for researchers who consider the analysis and classification of EEG signals. The task of diagnosing epilepsy is very important. There are studies to diagnose this disease using machine learning methods for early prediction of epileptic seizures. Typically, these methods include feature extraction, feature selection, and classification. According to the authors, this study is a systematic review of the literature on the feature selection process and classification performance. In general, this article is good.
However, there is a significant gap in the analysis of fuzzy classifiers. For these classifiers, the authors consider only [4] in which the accuracy is 96%results. But in papers published by IEEE, https://ieeexplore.ieee.org/document/9224666 and https://ieeexplore.ieee.org/document/9444205 have been considered fuzzy classifiers which have an accuracy of more than 99%. I suppose that these papers should be included in this review.

Author Response

Reviewer#3, Concern # 1: - This work is very interesting. This review may be useful for researchers who consider the analysis and classification of EEG signals. The task of diagnosing epilepsy is very important. There are studies to diagnose this disease using machine learning methods for early prediction of epileptic seizures. Typically, these methods include feature extraction, feature selection, and classification. According to the authors, this study is a systematic review of the literature on the feature selection process and classification performance. In general, this article is good.

However, there is a significant gap in the analysis of fuzzy classifiers. For these classifiers, the authors consider only [4] in which the accuracy is 96%results. But in papers published by IEEE, https://ieeexplore.ieee.org/document/9224666 and https://ieeexplore.ieee.org/document/9444205 have been considered fuzzy classifiers which have an accuracy of more than 99%. I suppose that these papers should be included in this review.

Author response: Thank you so much for this valuable suggestion. We have revised the manuscript and added suggested references as well as added 27 more references in the manuscript.

Author action: We have updated the classification table 4 by adding the suggested 2 references that are [68] and [69]. Moreover, 27 more references have been added in Tables 5, 6, 7, and 8.

Round 2

Reviewer 3 Report

The authors significantly extended and improve the review and also took into account my recommendation. I have no other comments and recomendations.